# Anti-Obesity Effect of Chitoglucan in High-Fat-Induced Obesity Mice

**DOI:** 10.3390/ijerph20010281

**Published:** 2022-12-24

**Authors:** Hyun-Jung Park, SunYoung Lee, Minsook Ye, Bong Hee Han, Hyun Soo Shim, Daehyuk Jang, Insop Shim

**Affiliations:** 1Department of Food Science and Biotechnology, Kyonggi University, 154-42, Gwanggyosan-ro, Youngtong-gu, Suwon 16227, Republic of Korea; 2Department of Physiology, College of Medicine, Kyung Hee University, 26 KyungHee-daero, Seoul 02447, Republic of Korea; 3Laboratory of Regenerative Medicine for Neurodegenerative Disease, Stand Up Therapeutics, Hannamdaero 98, Seoul 04418, Republic of Korea

**Keywords:** high fat induced mice, anti-obesity, *Flammulina velutipes*, chitoglucan, serum leptin, resistin

## Abstract

Background: Chitoglucan (CG) is a bioactive component obtained from *Flammulina velutipes Sing,* an edible mushroom, which is known to have an anti-obesity effect. However, its biological and hormonal activities in alleviating obesity through regulation of adipocyte-derived proteins have not been examined yet. Purpose: The present study aimed to investigate the anti-obesity effects of chitoglucan and its hormonal mechanisms in high-fat diet (HFD)-induced mice. Methods: The mice were fed either a normal diet (Normal group) or a high fat diet (HFD group) over 6 weeks. The HFD fed mice were administered with saline (HFD group), adipex (HFD + adipex group), chitoglucan 50, 150, or 300 mg/kg/day for 3 weeks (HFD + CG groups). The food consumption, body weight, fat contents, and the levels of serum leptin and resistin were assessed after treatment of chitoglucan. Results: the HFD produced a marked increase in body and fat weights after 6 weeks of feeding compared with the Normal group. Administration of chitoglucan for 3 weeks tended to reduce body weight and significantly decreased parametrical adipose tissues in HFD groups. The level of serum leptin in the HFD group was markedly higher than that in the Normal group, whereas the level of leptin in the chitoglucan treated groups was significantly decreased in comparison with the HFD group. In addition, the level of serum resistin in high-fat diet group tended to be more increased than Normal group. However, the serum resistin level was significantly reduced in HF diet groups after treatment with chitoglucan (50 mg/kg or 150 mg/kg). Conclusion: Collectively, these data suggest that chitoglucan from the *Flammulina velutipes* may be useful in the treatment of high diet-induced obesity and metabolic syndrome.

## 1. Introduction 

Obesity is a severe chronic syndrome with various causes characterized by over-deposition of fat in the human body. The cause of obesity is not yet clear, but accumulation of body fat due to imbalance in energy consumption is known as the main factor [1,2,3]. Abdominal obesity patients are often associated with pathological conditions such as X-syndrome and act as a powerful risk factor for early arteriosclerosis, ischemic cardiovascular disease and cerebrovascular disease [4,5,6]. The high-fat and high-sugar diet is related to cause of hyperlipidemia [7]. Kang et al. reported that high-fat-induced obesity altered cognition and anxiety related behavior. Treatment of obesity has a goal not only for weight loss but also for improving metabolism, which can be a predisposition to early cardiovascular disease. Disease theory of obesity involves motor nerves, autonomic nerves, and peripheral nervous systems [8,9,10] 

Adipose tissue can be divided into white adipose tissue (WAT) and brown adipose tissue (BAT). They are related to a complex network of endocrine organs. Specially, WAT is mainly responsible for the metabolic related diseases such as lipid oxidation, accumulation of fat, and diabetes, [11]. Leptin is one of the key mediators of lipid and glucose metabolic homeostasis [12]. The subcutaneous adipocytes release a higher amount of leptin than the visceral adipocytes [13]. Adipose tissue secrets leptin that controls food intake and energy expenditures [14], which was the first identified adipocytokine. Leptin is also known to regulate energy homeostasis and the immune system [14]. Leptin is related to a loss of social anhedonia [15]. It was reported that leptin stimulated proliferation of gastric cancer cells via regulation of the Janus kinase (JAK)-signal transducer and activator of transcription (STAT) and the mitogen activated protein kinase (MEK) signaling pathway [16]. A clinical study reported high correlations of leptin resistance with young obese women [17]. Resistin is also adipose tissue-derived signaling cysteine-rich protein [14]. The pro-inflammatory cytokines are known to play a key role in the pathogenesis of diabetes where resistin may be responsible for these inflammatory processes and insulin resistance [11]. It has been shown that resistin is related to knee osteoarthritis [18] and acute pancreatitis [19]. The increase of resistin serum level was reported to link to ovarian cancer [20]. In a human study, resistin elevation was observed in the adipose tissue of obese patients [21]. However, intermittent energy restriction reduced inflammatory cytokines and chemokines (interleukin (IL)-1beta, interferone (IFN)-gamma, tumor necrosis factor (TNF)-alpha, IL-6, IL-8, IL-12, IL-23) and adipokines (leptin, resistin, adiponectin, adipsin) [22]. Two hormones, leptin and resistin, are best known as adipose tissue-specific secretory factors, playing a critical role in obesity and diabetes. 

*Flammulina velutipes*, an edible mushroom, is known to have an antioxidant effect on lipid oxidation [23]. Extract of *Flammulina velutipes* showed immune-enhancing properties via regulation of cytokine production and reactive oxygen species production [24]. Chitoglucan (CG) is a bioactive component obtained from *Flammulina velutipes Sing* [23]. Chitoglucan is well known for reducing neutral fat and blood pressure [25]. In addition, beta-glucan, one of the main components of chitoglucan, has been shown to exhibit anti-obesity effects by inducing improvement of lipid metabolism [26,27,28]. Although in vitro, chitoglucan was suggested to have anti-adipogenic effects, its biological and hormonal activities in alleviating obesity through regulation of adipocyte-derived proteins have not been examined yet. 

The present study evaluated the anti-obesity activity of chitoglucan through examination of changes in body weight, food consumption, and total fat weight in mice. Additionally, hormonal mechanisms underlying the anti-obesity effect of chitoglucan were elucidated through the analysis of serum levels of leptin and resistin.

## 2. Methods and Materials

### 2.1. Animals and Diets

All the experiments were approved by the Kyung Hee University institutional animal care (KHUASP(SE)-13-041) and in accordance with the US National Institutes of Health “Guide for the Care and Use of Laboratory Animals” (NIH Publication number 80–23, revised 1996). Seven-week-old male C57BL/6J mice were purchased from Orient Corp (Seongnam-si, Republic of Korea). The mice were fed a standard chow diet or a high-fat diet for 6 weeks. The temperature was controlled at 20–25 °C (40–45% humidity) with a 12:12 light–dark cycle. The diets contained either a normal amount of fat (Rident pellet, Samyang Corp, Seongnam-si, Republic of Korea), or high amount of fat (Dyets Inc., Bethlehem, PA, USA, Table 1). The animals were divided into 6 groups:Normal group: Normal diet groupHFD group: High fat diet and saline (1 mL/kg, P.O.)HFD + adipex: High fat dietand adipex (0.3 mg/kg, P.O.)HFD + CG 50 mg/kg: High fat diet and chitoglucan 50 mg/kg treatment (50 mg/kg, P.O)HFD + CG 150 mg/kg: High fat diet and chitoglucan 150 mg/kg treatment (150 mg/kg, P.O)HFD + CG 300 mg/kg: High fat diet and chitoglucan 300 mg/kg treatment (300 mg/kg, P.O)

### 2.2. Chitoglucan and Adipex

Chitoglucan was provided by SunJin Clover (Incheon, Republic of Korea). For the positive control, adipex (phentermine) was purchased from ADIPEX-P^®^ (37.5 mg phentermine hydrochloride tablets containing 30 mg phentermine base). From the 7th week to the 10th week, mice received ADIPEX-P^®^, chitoglucan, or saline at 1 mL/kg. 

### 2.3. Measurement of Food Consumption, Body Weight, and Regional Fat Weight

The food containers were daily given to animals with fresh food. The food consumption was measured by the amount of their food in their home, which was calculated by subtracting the residual food quantity. For the body weight, all mice were weighed manually the same day each week. For the regional fat weight, the mice were anesthetized with an intraperitoneal injection of sodium pentobarbital anesthesia (50 mg/kg). The perirenal fat, mesenteric fat, and epididymal fat of each mouse were isolated. Then, the total weight of perirenal fat, mesenteric fat, and epididymal fat were measured. 

### 2.4. Measurement of Serum Level of Leptin and Resistin 

Blood was drawn from the heart under sodium pentobarbital anesthesia (50 mg/kg, i.p.) and centrifuged (13, 000× *g* for 10 min at 4 °C). Serum samples were frozen at −80 °C. The level of serum leptin (R&D system Inc., Santa Clara, CA, USA) was determined by using a mouse ELISA kit according to the manufacturer’s instructions. Briefly, all samples were examined in duplicate per mouse. A total of 100 µL of sample in reagent diluent was add to each well. The plate was incubated for 2 h at room temperature. Then the plate was washed three times with washing solution and 100 µL of detection antibody was added to each well. Then the plate was washed three times with washing solution. Then, 100 µL streptavidin-HRP solution was added to each well. Then the plate was washed three times with washing solution. Into the empty well was added 50 µL of stop solution. Using the microplate reader, the optical density was examined at 450 nm.

The level of serum resistin (Abcam, Cambridge, UK) was determined by using a mouse ELISA kit accordingly the manufacturer’s instructions. Briefly, all samples were examined in duplicate per mouse. A total of 50 µL of sample and 50 µL of antibody cocktail in reagent diluent was added to each well. The plate was incubated for 1 h at room temperature on a plate shaker set to 400 rpm. Then plate was washed three times with washing solution and 100 µL of detection antibody was added to each well. Then the plate was washed three times with washing solution. Then, 100 µL TMB substrate solution was added to each well on a plate shaker set to 400 rpm. Then the plate was washed three times with washing solution. To the empty well was added 100 µL of stop solution. Using the microplate reader, the optical density was examined at 450 nm.

Data were analyzed in duplicate in one assay and intra- and inter-assay variation was below 10%.

### 2.5. Statistical Analysis 

All of the results were showed as the mean ± S.E.M. Statistical analysis was analyzed with SPSS 25.0 software (SPSS 25 Inc., Chicago, IL, USA) using one-way ANOVA and LSD post hoc test. *p*-value < 0.05 was considered statistically significant. Graph generations were made with GraphPad Prism 6.0 software.

## 3. Results

### 3.1. Body Weight 

The body weight of all mice was examined once a week. As shown in Figure 1, the HFD group gained more body weight than the Normal group (*p* < 0.001). However, after treatment of CG, body weight in the HFD + CG groups tended to be more decreased than the HFD group. Therefore, it was shown that chitoglucan has the potential to reduce weight gain. 

### 3.2. Food Consumption

The food consumption of all mice was examined once a week. As shown in Figure 2, regarding the food consumption for 6 weeks, there was not a significant difference among groups. However, after treatment (seventh week), of CG, the food consumption of in the HFD + CG groups tended to be more decreased than the HFD group. Therefore, these results showed that chitoglucan has the potential to reduce weight gain via regulation of food intake. 

### 3.3. Fat Storage

The total weight of perirenal fat, mesenteric fat, and epididymal fat were measured. The total fat weights were higher in the HFD-fed groups than in the Normal group (*p* < 0.001, Figure 3). The total fat weight is almost four times higher in HFD group than that of the Normal group. However, chitoglucan-treated groups significantly reduced the total fat weight compared with the HFD group (*p* < 0.001). 

### 3.4. Enzyme Linked Immunosorbent Assay (ELISA)

#### 3.4.1. Leptin

The serum leptin level was examined using the ELISA kit. As shown in Figure 4, the level of serum leptin in the HFD group was markedly higher than that in Normal group. However, serum leptin level of chitoglucan-treated groups was significantly reduced compared to the HFD group (*p* < 0.05).

#### 3.4.2. Resistin

The serum resistin level was examined using the ELISA kit. The effect of CG on the serum resistin level is shown in Figure 5. The serum resistin level in the HFD fed group tended to be more increased than Normal group. However, the serum resistin level of the CG (50 mg/kg or 150 mg/kg)-treated group was markedly decreased compared with the HFD group (*p* < 0.01).

## 4. Discussion

The present study showed that body weight, food intake, and serum leptin level in the HFD group were significantly increased compared to the Normal group. However, after treatment with chitoglucan, level of serum leptin was significantly decreased compared to the HFD group. The serum resistin level in the HFD fed group tended to be more increased than Normal group. However, after treatment of CG (50 mg/kg or 150 mg/kg), the serum resistin level was markedly decreased compared with the HFD group. These data proved that the chitoglucan from the *Flammulina velutipes* may be useful in the prevention of high-fat diet-induced obesity.

Over 1 billion people worldwide are obese. Obesity affects the liver, kidney, heart, joints, muscles, and reproductive system. Obesity is recognized as a disease that threatens public health as it is regarded as a central pathological mechanism of metabolic syndrome, which is characterized by high blood pressure, diabetes, and abnormal lipid metabolism [17,29]. The impact of obesity on global public health has grown in recent years. The basic pathogenesis of obesity includes either up-regulation of appetite or down-regulation of calorie utilization by governing physical activity [30]. Additionally, obesity is related with metabolic disorders, inducing fatty liver and lipid metabolism disorders. Adipocytes (fat cells) are associated with energy homeostasis and lipid pathway signaling.

Leptin, a regulator of energy homeostasis, is an adipocyte-derived protein that functions as an adipostat to sense and regulate body energy stores. High-fat diet-induced obesity was related either to the failure of leptin synthesis and secretion [31]. The change of leptin level alters food intake and control of energy expenditures [14]. We observed that serum leptin was increased in the high-fat-diet group, but treatment with chitoglucan significantly reduced leptin level. Consistent with the result of leptin level, resistin was significantly reduced after treatment of chitoglucan. Parallel to hormonal changes of leptin and resistin, body weight, and the weights of regional epididymal, perirenal, and peritoneal fat mass were markedly reduced in chitoglucan-treated groups. The present results demonstrated that the concentrations of leptin were highly correlated with final body weight and three reginal fats, which was most remarkable. Consistent with the current results, trials with human subjects have previously shown that intake of mushroom chitosan controls total serum cholesterol and neutral fats, and decreases visceral fat [32]. Furthermore, as a mechanism for specific visceral fat reduction effect, mushroom chitosan is reported to have relatively high affinity for β-adrenergic receptors [32]. The results of the present study suggest that chitoglucan may contribute to the decrease of body fat and regulating leptin secretion. 

Parallel to leptin level, resistin was significantly reduced after treatment of chitoglucan. Resistin was known as an adipocyte hormone that plays a key role in the regulation of glucose metabolism [33]. Serum resistin was positively correlated with changes of body adipose mass [33]. In human studies, resistin is expressed in adipocytes, and obese humans showed resistin elevation in the adipose tissue [21]. The previous study has shown that resistin was increased in diet-induced obesity as well as in genetic models of obesity and insulin resistance [11]. Plasma concentrations of resistin were higher in genetic and diet-induced obese mice with insulin resistance [34]. However, gene expression of resistin was markedly down-regulated by treatment with anti-diabetic drugs called thiazolidinediones that improve target-tissue sensitivity to insulin [11]. We observed that the serum resistin level and total fat weight in the HFD-fed group tended to be more increased than the Normal group, but treatment of chitoglucan reduced resistin and the weights of regional epididymal, perirenal, and peritoneal fat mass. 

It was previously shown that polysaccharide from *Flammulina velutipes* attenuated lipid metabolism in high-fat diet fed mice [35]. The previous study reported that active compound of *Flammulina velutipes* decreased HDL in the high-fat diet fed hamsters [36]. The concentrations of leptin and resistin were highly correlated with final body weight and three reginal fats, which was most remarkable. Chitoglucan is a bioactive component consisting of chitosan, beta glucans, fatty acids, and fiber. It is known that two major components, chitosan and fatty acids, have the most biological activity of chitoglucan. Anti-obesity effects of chitoglucan may be the result of a synergetic effect of its various components including chitosan, which inhibits lipid absorption, and fatty acid complexes, which decompose body fat. 

Taken together, these data suggest that chitoglucan may be helpful in the prevention of diet-induced obesity and metabolic syndrome via regulation of resistin expression. In light of such limitations, we did not further analyze other physiological obesity markers such as orexigenic peptides such as neuropeptide Y, ghrelin, CCK, or orexin in the present study. More precise mechanisms of chitoglucan using these biomarkers should be further examined.

## 5. Conclusions

In summary, treatment with chitoglucan from *Flammulina velutipes* reduced body weight, food intake, and body fat induced by high fat diet via regulation of the level of serum resistin and leptin. Collectively, these data suggested that the chitoglucan from the *Flammulina velutipes* may be a helpful compound for the alleviation of obesity (Figure 6).

## Figures and Tables

**Figure 1 ijerph-20-00281-f001:**
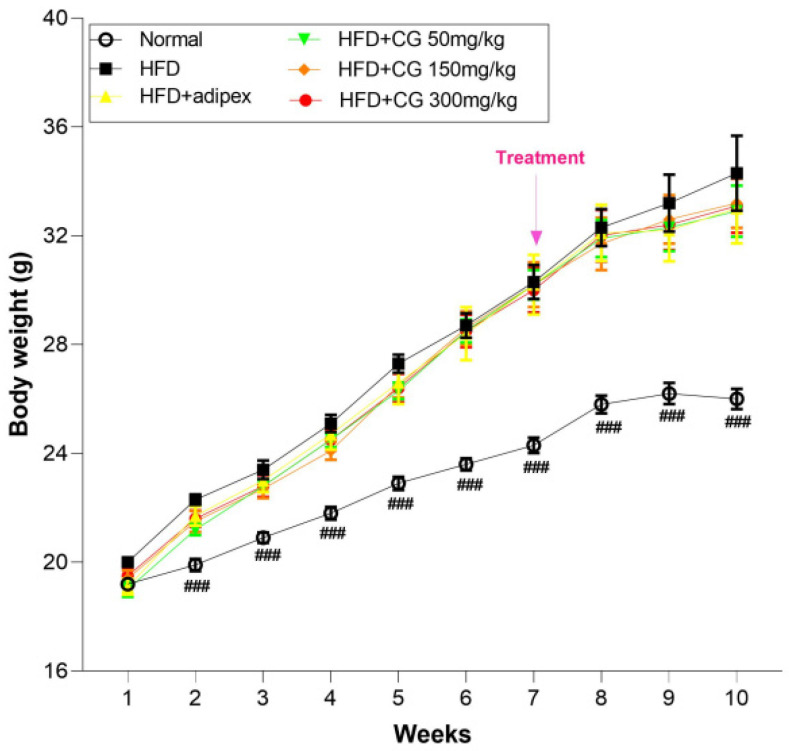
Effect of chitoglucan on body weight. Data represent means ± S.E.M. ### *p* < 0.001 compared with HFD group.

**Figure 2 ijerph-20-00281-f002:**
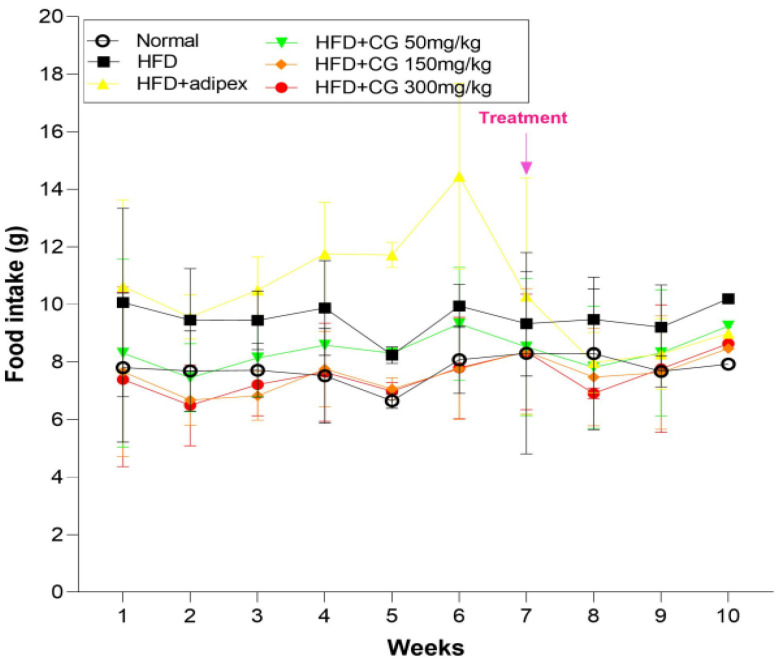
Change in food intake. Data represent means ± S.E.M.

**Figure 3 ijerph-20-00281-f003:**
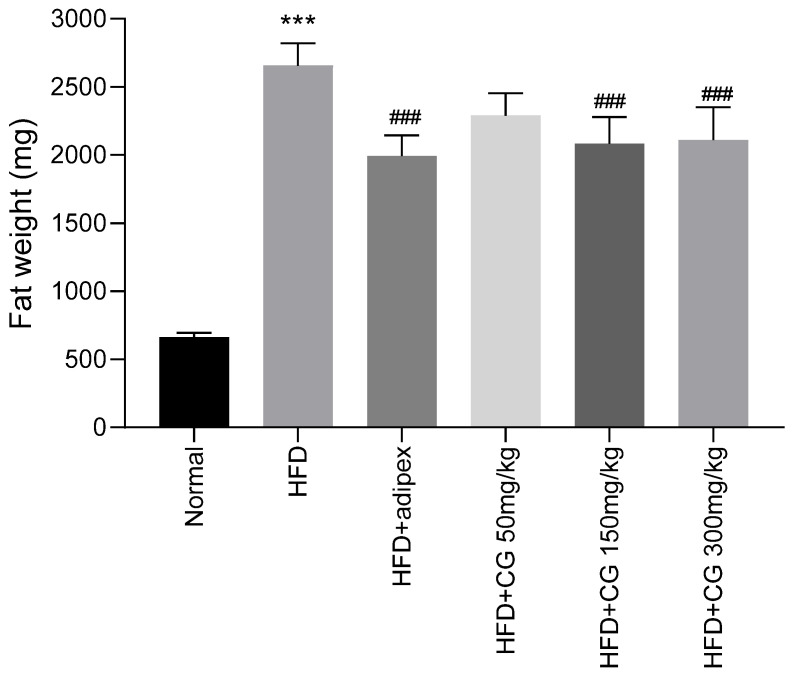
Effect of chitoglucan on fat weight. Data represent means ± S.E.M. *** *p* < 0.001 compared with Normal group, ### *p* < 0.001 compared with HFD group.

**Figure 4 ijerph-20-00281-f004:**
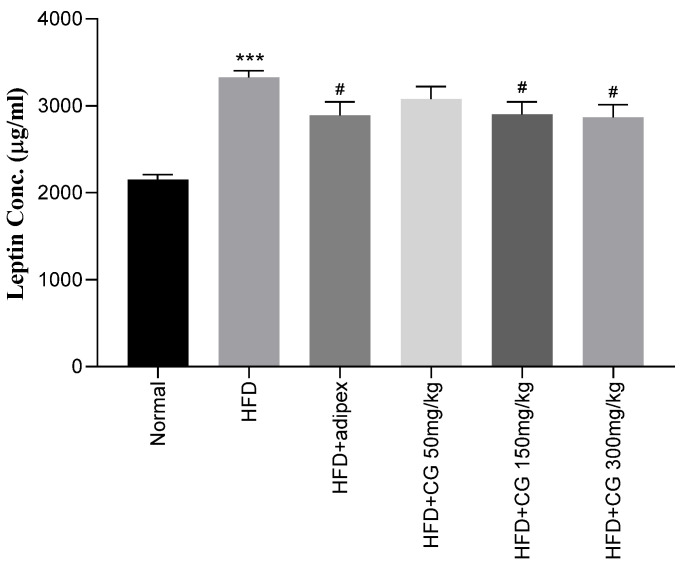
Effect of chitoglucan on serum level of leptin. Data represent means ± S.E.M. *** *p* < 0.001 compared to Normal group, **#**
*p* < 0.05 compared to HFD group.

**Figure 5 ijerph-20-00281-f005:**
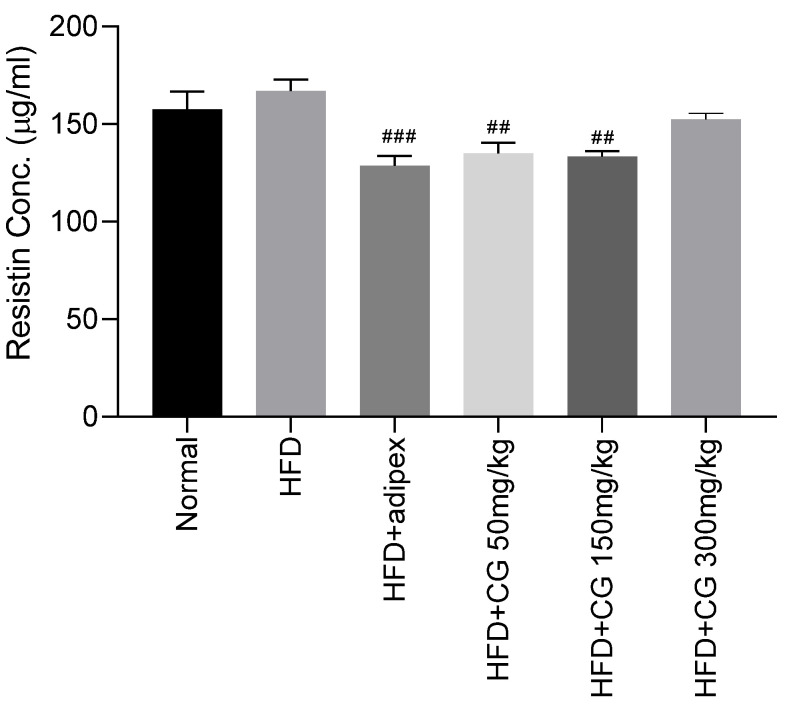
The results of serum resistin level of the mice. Data represent means ± S.E.M. **##** *p* < 0.01, **###** *p* < 0.001 compared to HFD group.

**Figure 6 ijerph-20-00281-f006:**
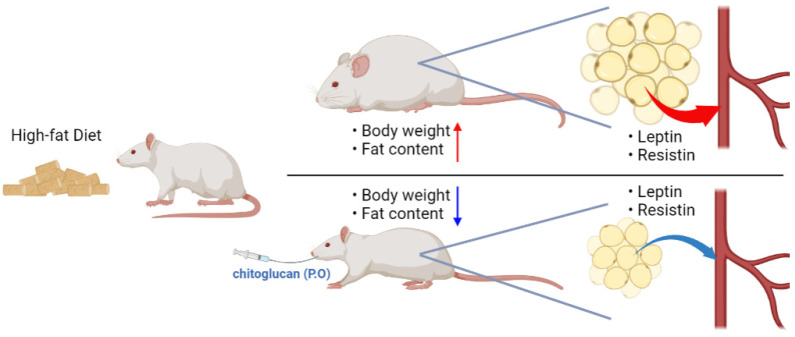
Summary key findings.

**Table 1 ijerph-20-00281-t001:** Composition of the experimental diets.

Ingredient	Normal Diet ^(1)^	Hugh Fat Diet ^(2)^
Casein	200	200
DL-Methionine	3	3
Corn Starch	150	150
Sucrose	500	345
Cellulose	50	50
Corn Oil	50	-
Beer Tallow	-	205
Salt Mixture	35	35
Vitamin Mixture	10	10
Choline bitartrate	2	2
Fat % (Calories)	11.7	40.0

^(1)^ Normal fat diet: AIN-76A diet #100000 (Dyets Inc., Bethlehem, PA, USA); ^(2)^ High fat diet: AIN-76 diet #100476 (Dyets Inc., Bethlehem, PA, USA).

## Data Availability

Data sharing not applicable.

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
