# Peer review of "Anti-Obesity Effect of Chitoglucan in High-Fat-Induced Obesity Mice"

_ijerph, 2022, doi:10.3390/ijerph20010281_

Round 1

Reviewer 1 Report

The research is engaged in a relevant topic providing remarkable results, namy pointing out that chitoglucan from Flammulina velutipes might be a useful tool in terms of prevention of diet-induced obesity and metabolic syndrome. My questions are as follows:

- Have you investigated the structure and quality of the applied chitoglucan?

- Do you have any experimental data on the efficiency of the applied agent in comparison with other chitoglucan batches?

- The level of serum leptin in case of chitoglucan treated mice groups was significantly decreased compared to the HFD group. Could you underpin this observation by means of other experimental evidences?

- The serum resistin in chitoglucan diet group displayed a decreased level compared to the HFD group. What is the observation in comparison with the control group?

- Have you implied additional methods to support your findings? Have you involved studies on other physiological obesity-markers?

Author Response

Author's Reply to the Review Report (Reviewer 1)

Anti-obesity effect of Chitoglucan in high-fat induced obesity mice (ijerph-1945201)

The research is engaged in a relevant topic providing remarkable results, namy pointing out that chitoglucan from Flammulina velutipes might be a useful tool in terms of prevention of diet-induced obesity and metabolic syndrome. My questions are as follows:

- Have you investigated the structure and quality of the applied chitoglucan?

Response: Thank you for reviewer’s comment. We did not investigate the structure and quality of the chitoglucan in the present study. However, the structure of Chitoglucan used in this study is well characterized and purified by an expert of the SunJIn Clover (Co.) who provided the product for this study. Its quality is well controlled. Standard commercial products of chitoglucan are also available in several professional companies.

- Do you have any experimental data on the efficiency of the applied agent in comparison with other chitoglucan batches?

Response: Chitoglucan is a bioactive component consisting of chitosan, beta glucans, fatty acids, and fibers etc., and two major components, chitosan and fatty acids are known to have the most of biological activity of chitoglucan. We did not directly compare our product with other chitoglucan batches, but we assume that chitoglucan products may have a similar anti-obesity effect observed in the present study since most of chitoglucan extracts contain chitosan and fatty acids.

- The level of serum leptin in case of chitoglucan treated mice groups was significantly decreased compared to the HFD group. Could you underpin this observation by means of other experimental evidences?

Response: As reviewer’s comment, we included more discussions on leptin level regarding with other experimental results (See also, Lines 231-241).

Leptin, a regulator of energy homeostasis, is an adipocyte-derived protein that functions as an adipostat to sense and regulate body energy stores. We observed that serum leptin was increased in high fat-diet group, but treatment with chitoglucan significantly reduced leptine level. Consistent with result of leptin level, resistin was significantly reduced after treatment of chitoglucan. Parallel to hormonal changes of leptin and resistin, body weight, food consumption, the weights of regional epididymal, perirenal, and peritoneal fat mass were markedly reduced in chitoglucan-treated groups. The concentrations of leptin were highly correlated with final body weight and three reginal fats, which was most remarkable.

- The serum resistin in chitoglucan diet group displayed a decreased level compared to the HFD group. What is the observation in comparison with the control group?

Response: As reviewer’s comment, chitoglucan treated group displayed a decreased level of serum resistin compared with the HFD group, but there was not significant difference between two groups. The serum resistin level in HFD fed group tended to be more increased than Normal group. (See also, Lines 206-207)

- Have you implied additional methods to support your findings? Have you involved studies on other physiological obesity-markers?

Response: Thank you for your good suggestions and comments. We did not further analyze other physiological obesity-markers such as orexigenic peptides such as neuropeptide Y, ghrelin, CCK or orexin in the present study. However, we are planning to examine more precise mechanisms of chitoglucan using these biomarkers. Now, we add the other studies’ findings and limitation of this present study in the discussion section. (See also, Lines 272-276)

Reviewer 2 Report

Title: Anti-obesity effect of Chitoglucan in high-fat induced obesity mice

1.       Please check the format of abstract.

2.       In abstract, “Background Flammulina velutipes is known to be effective in suppressing lipid oxidation and cancer in white mouse brain tissue by active oxygen. Purpose: Present study aimed to prove  the possible anti-obesity effects of chitoglucan” This is unacceptable. Please explain a good rationale for your intention to investigate the anti-obesity properties of chitoglucan.

3.       Please double-check the manuscript's abbreviation. Please abbreviate the full names if you have previously stated them.

4.       Please provide the certificate of approval number (COA. No.) of the animal study

5.       Why did you use 50, 150 and 300 mg/kg chitoglucan? What is the rationale behind that?

6.       Why did all mice suddenly avoid eating at week 10?

7.       Please draw Table 1 by yourself in order to make this seems professional.

8.       Why did you choose the seventh week to begin treating mice?

9.       Although you bought chitoglucan, please provide some chrematistics of this chemical, such as structure, preparation and purity.

10.   Before drawing any conclusions from Figs. 1 and 2, statistical analysis must be performed.

11.   Fig 3, HFD+50 showed ###. This is impossible. The mean value seems very close to HFD.

12.   Fig 4, HFD+150 showed #, while HFD+50 showed ###. How is this possible? Because HFD+50 showed less effectiveness compared to HFD+150. This make me concern for the whole manuscript. If you have chance to revise the manuscript, please provide the raw data or the calculation how did you perform statistical analysis.

13.   µg not ug

14.   S.E.M or SEM?

15.   Why did you used LSD post hoc test? Why not Duncan test?

16.   Line 197: The serum leptin level was examined using the ELISA kit?

17.   Line 243: “reduces the adipose tissue in the body” How can you conclude this without any tests?

18.   The discussion is poor. The authors did not discuss their data in deep. Most text are just previous finding or general biochemistry knowledge.

19.   Fig 7. The conclusion is inappropriate. According to Fig7, I can summarize that you do not need to eat chitoglucan; only consume less calories.

20.   Format of Refs was wrong. Please read and follow the guideline.

Author Response

Author's Reply to the Review Report (Reviewer 2)

Please check the format of abstract.

Response: As reviewer’s comment, we changed the background in abstract section.

  1. In abstract, “Background Frameline velutipesis known to be effective in suppressing lipid oxidation and cancer in white mouse brain tissue by active oxygen. Purpose: Present study aimed to prove the possible anti-obesity effects of chitoglucan” This is unacceptable. Please explain a good rationale for your intention to investigate the anti-obesity properties of chitoglucan.

Response: As reviewer’s comment, we clearly state the rationale and purpose of the study in abstract section as follows: (See also, lines 20-25)

Background Chitoglucan (CG) is a bioactive component obtained from Flammulina velutipes Sing, an edible mushroom, which is known to have an anti-obesity efficacy. However, its biological and hormonal activities in alleviating obesity through regulation of adipocyte-derived proteins have not been examined yet. Purpose: The present study aimed to investigate the anti-obesity effects of chitoglucan and its hormonal mechanisms in high-fat diet (HFD)-induced mice

  1. Please double-check the manuscript's abbreviation. Please abbreviate the full names if you have previously stated them.

Response: As reviewer’s comment, we check all abbreviation in whole manuscript.

(See also, lines 65-66, line 75)

  1. Please provide the certificate of approval number (COA. No.) of the animal study

Response: As reviewer’s comment, we add the certificate of approval number.

(See also, line 97)

  1. Why did you use 50, 150 and 300 mg/kg chitoglucan? What is the rationale behind that?

Response: Several studies have reported a variety of biological activities of chitoglucan and chitosan, its main component extracted from Flammulina velutipes. In particular, anti-obesity effects of chitosan prepared from Flammulina velutipes were demonstrated using dosages from 100mg/kg to 1000mg/kg in animal studies. According to the previous study, we selected dosage rangeing from 50, 150 to 300 mg/kg of chitoglucan in the present study [1].

  1. Why did all mice suddenly avoid eating at week 10?

Response: The mice were fed either a normal diet or a high fat diet over 6 weeks. Thereafter, the HFD-fed mice received chitoglucan for 3 more weeks. On 10th week, mice were sacrificed for the analysis of hormonal changes of leptin and resistin.

  1. Please draw Table 1 by yourself in order to make this seems professional.

Response: As reviewer’s comment, we revised the Table 1 in the manuscript.

  1. Why did you choose the seventh week to begin treating mice?

Response: According to previous studies, we chose the seventh week to begin treating mice [2, 3].

  1. Although you bought chitoglucan, please provide some chrematistics of this chemical, such as structure, preparation and purity.

Response: Thank you for reviewer’s comment.

Chitoglucan is a bioactive component extracted from the edible enokitake Mushroom. It is obtained from water extraction of the fruiting body of enokitake Mushroom, followed by alkali addition to the extracted liquid. It is known that the resulting extract consist of chitosan, beta glucans, fatty acids, and glyconjugates, ect. Chitoglucan used in the present study was provided a professor company (Sun Jin Clover, Co), and the standard commercial products are also available in the several companies. 

  1. Before drawing any conclusions from Figs. 1 and 2, statistical analysis must be performed.

Response: As reviewer’s comment, we performed a statistical analysis of our data using one-way ANOVA, followed by post-hoc test. We have modified Figure 1 and 2.  

  1. Fig 3, HFD+50 showed ###. This is impossible. The mean value seems very close to HFD.

Response: As reviewer’s comment, we checked stats again and now we revised the figure 3 in the manuscript.

  1. Fig 4, HFD+150 showed #, while HFD+50 showed ###. How is this possible? Because HFD+50 showed less effectiveness compared to HFD+150. This make me concern for the whole manuscript. If you have chance to revise the manuscript, please provide the raw data or the calculation how did you perform statistical analysis.

Response: According to reviewer’s comment, we checked out whole raw data and stats. (See also, Figure 4). Now we provide raw data of leptin level of serum. Statistical analysis was analyzed with SPSS 25.0 software (SPSS 25 Inc., Chicago, IL) using one-way ANOVA and LSD post hoc test. P-value <0.05 was considered statistically significant. Graph generations were followed with GraphPad Prism 6.0 software.

  1. µg not ug

Response: As reviewer’s comment, we have corrected errors.  

  1. S.E.M or SEM?

Response: We have corrected “SEM to S.E.M” in whole manuscript.

  1. Why did you used LSD post hoc test? Why not Duncan test?

Response: The reviewer brings up an important point. LSD is the best for all-possible pairwise comparisons when sample sizes are unequal, or confidence intervals are needed, very good even with equal samples sizes without confidence intervals. However, Duncan use for comparing one sample (“control”) to each of the others, but not comparing the others to each other

  1. Line 197: The serum leptin level was examined using the ELISA kit?

Response: As reviewer’s comment, we revised that. (See also, Line 196)

  1. Line 243: “reduces the adipose tissue in the body” How can you conclude this without any tests?

Response: As reviewer’s comment, now we revised this sentence (see also, Lines 228-230)

  1. The discussion is poor. The authors did not discuss their data in deep. Most text are just previous finding or general biochemistry knowledge.

Response: As reviewer’s comment, we discuss our data in deep (See also, Lines 228-276).

  1. Fig 7. The conclusion is inappropriate. According to Fig7, I can summarize that you do not need to eat chitoglucan; only consume less calories.

Response: As reviewer’s comment, we have corrected conclusion and the Fig. 7.

  1. Format of Refs was wrong. Please read and follow the guideline.

Response: As reviewer’s comment, we have corrected reference format and followed Journal guideline.

  1. J. Oleo Sci. 67, (2) 245-254 (2018) Improvement of Diet-induced Obesity by Ingestion of Mushroom Chitosan Prepared from Flammulina velutipes
  2. Park, H. J.; Jung, E.; Shim, I., Berberine for Appetite Suppressant and Prevention of Obesity. BioMed research international 2020, 2020, 3891806.
  3. Park, H. J.; Kim, J. H.; Shim, I., Anti-obesity Effects of Ginsenosides in High-Fat Diet-Fed Rats. Chinese journal of integrative medicine 2019, 25, (12), 895-901.

Round 2

Reviewer 1 Report

The comments and amendments are satisfactory, and provide sufficient ground for the drawn conclusions. The methodolody part is extended in the right direction.

Reviewer 2 Report

The authors revised their work in response to the suggestions.